# Comparative efficacy of tenofovir and entecavir in nucleos(t)ide analogue-naive chronic hepatitis B: A systematic review and meta-analysis

**Mao-bing Chen** [1]☯*, **Hua Wang**[2]☯, **Qi-han Zheng**[1], **Xu-wen Zheng**[1], **Jin-nuo Fan**[1], **Yun-long Ding**[3], **Jia-li Niu**[4]

1 Department of Emergency, Wujin People Hospital Affiliated with Jiangsu University and Wujin Clinical College of Xuzhou Medical University, Changzhou, Jiangsu, P. R. China, 2 Department of ICU, Wujin People Hospital Affiliated with Jiangsu University and Wujin Clinical College of Xuzhou Medical University, Changzhou, Jiangsu, P. R. China, 3 Department of Neurology, Jingjiang People Hospital, Seventh Affiliated Hospital of Yangzhou University, Jingjiang, Jiangsu, P. R. China, 4 Department of Clinical Pharmacy, JingJiang People Hospital, Seventh Affliated Hospital of Yangzhou University, Jingjiang, Jiangsu, P. R. China

☯ These authors contributed equally to this work.
* 554118854@qq.com

**Data Availability Statement:** All relevant data are within the manuscript and its Supporting Information files.

## Abstract

### Objective

To compare the efficacy of tenofovir and entecavir in nucleos(t)ide analogue-naive chronic hepatitis B.

### Methods

The Web of Science, PubMed, Cochrane Library, EMBASE, Clinical Trials and China National Knowledge Infrastructure(CNKI) databases were electronically searched to collect randomized controlled trials (RCTs) regarding the comparison between tenofovir and entecavir in nucleos(t)ide analogue-naive chronic hepatitis B (CHB) since the date of database inception to July 2019. Two researchers independently screened and evaluated the obtained studies and extracted the outcome indexes. RevMan 5.3 software was used for the meta-analysis.

### Results

Early on, tenofovir had a greater ability to inhibit the hepatitis B virus, $I^2 = 0\%$ [RR = 1.08, 95% CI (1.03, 1.13), $P<0.01$] (96 weeks). Entecavir can normalize the ALT levels earlier, $I^2 = 0\%$ [RR = 0.87, 95% CI (0.77, 0.98), $P = 0.02$] (48 weeks). However, there was no statistically significant difference between TDF and ETV at 144 weeks. Tenofovir was as effective as entecavir in terms of HBeAg clearance and HBeAg seroconversion, $I^2 = 0\%$ [RR = 1.05, 95% CI (0.68, 1.62), $P = 0.82$]; $I^2 = 69\%$ [RR = 0.93, 95% CI (0.54, 1.61), $P = 0.80$]. The difference in the incidence of elevated creatine kinase levels was not statistically significant $I^2 = 0\%$ [RR = 0.66, 95% CI (0.27, 1.60), $P = 0.35$].

**Funding:** This work was funded by the Guidance plan for social development of Changzhou Municipal Science and Technology (CE20175008), Changzhou City, Jiangsu Province, China, to MC. The funder had no role in the study design, data collection, data analysis, data interpretation, writing of the report, decision to publish, or preparation of the manuscript.

**Competing interests:** The authors have declared that no competing interests exist.

**Abbreviations:** CNKI, China National Knowledge Infrastructure; RCTs, randomized controlled trials; CHB, chronic hepatitis B; TDF, tenofovir disoproxil fumarate; ETV, entecavir; PROSPERO, international prospective register of systematic reviews; CK, creatine kinase; MD, mean difference; CI, confidence interval; HCC, hepatocellular carcinoma; LLQ, lower limit of quantitation.

## Conclusions

Tenofovir and entecavir were equally effective in the treatment of patients with nucleos(t)ide analogue-naive chronic hepatitis B. In addition, TDF has an advantage in the incidence of hepatocellular carcinoma. Additional RCTs and a large-sample prospective cohort study should be performed.

## 1. Introduction

Chronic hepatitis B (CHB) is indicated when there is continued positivity for the hepatitis B virus (HBV) and the course of the disease exceeds half a year or the date of infection is not known, with clinical manifestations of the disease[1]. The clinical manifestations are asthenia, fear of food, nausea, abdominal distension, liver pain and other symptoms[2, 3]. The liver is large, moderately hard and tender. Severe cases can be accompanied by symptoms of chronic liver disease, spider nevus, liver palm, and abnormal liver function[3, 4]. According to the World Health Organization report, more than 2 billion people have been infected with HBV worldwide, and approximately 240 million of them are chronically infected[5]. The current CHB guidelines recommend tenofovir disoproxil fumarate (TDF) or entecavir (ETV) for the treatment of CHB. As first-line drugs for CHB treatment, they have the common advantages of high antiviral efficacy, good tolerance and excellent genetic barrier, and it is not easy to develop drug resistance to them[6].

Patients with CHB need long-term antiviral treatment. Currently, there is no clear drug withdrawal guideline for antiviral treatment[7]. It is generally believed that antiviral drugs require long-term or even lifelong oral administration to achieve the goal of controlling CHB [8]. Patients often have questions about whether TDF or ETV is more appropriate at the time of initial treatment or in the early stages of CHB and whether TDF is better than ETV in terms of efficacy and safety[9]. In this study, the efficacy and safety of TDF and ETV in CHB patients were compared to provide a basis for patients to choose the more appropriate antiviral drug.

Prior to this study, there were similar systematic analysis articles, but at that time, there were few reliable randomized controlled trials (RCTs). In the past two years, relevant RCT studies have been published in journals. This study collected and analyzed those studies.

## 2. Methods

### 2.1. Design and registration

A meta-analysis was conducted to evaluate the effectiveness of TDF and ETV in nucleos(t)ide analogue-naive CHB. This protocol was registered in the international prospective register of systematic reviews (PROSPERO), registration number: CRD42019134194 (https://www.crd.york.ac.uk/PROSPERO). No ethics approval is required because this study used data that were already in the public domain.

### 2.2. Study selection

**2.2.1. Study type.** The studies in this analysis were RCTs.

**2.2.2. Study subjects.** Patients with definite CHB and no prior experience with nucleos(t)ide analogue therapy were included. The following patients were excluded: patients who were infected with HIV or other hepato-tropic viruses; those who had drug-induced liver diseases,

alcoholic liver disease or autoimmune liver diseases, tumors, serious complications in the heart, kidney, brain and other organs; and patients who were in pregnant or lactating.

**2.2.3. Intervention.** In the TDF group, the enrolled patients were given the conventional dosage of tenofovir 300 mg/day orally. In the ETV group, the enrolled patients were given the conventional dosage of entecavir 0.5 g/day orally.

**2.2.4. Outcome indicator.** The following outcomes were assessed and compared between the TDF and ETV groups: (1) differences in the probability of normalized ALT indicators, (2) differences in the probability of HBV-DNA negative results (undetectable), (3) differences in the probability of hepatitis E antigen clearance (HBeAg clearance), (4) differences in the probability of HBeAg seroconversion, and (5) differences in the probability of increased creatine kinase (CK) levels.

**2.2.5. Exclusion criteria.** Studies with data that could not be extracted or utilized, studies with animal experiments; and literature reviews were excluded.

## 2.3. Data sources and searches

We searched English and Chinese language publications through June 2019 using the following databases: Web of Science, PubMed, the Cochrane Library, EMBASE, Clinical Trials and the CNKI. The search terms included "Tenofovir", "Entecavir", and "Hepatitis B, Chronic". In Fig 1, we use the PubMed database as an example.

## 2.4. Study screening, data extraction and assessment of the risk of bias

Data were collected independently by two researchers. The unqualified studies were eliminated, and the qualified ones were selected after reading the title, abstract and full text. Then, the research data were extracted and checked, and disagreements were discussed or a decision was made by the authors. The extracted data included the following: 1. basic information of the study, including title, author and year of publication; 2. characteristics of the included study, consisting of the study duration, the sample size of the test group and the control group, and the intervention measures; 3. The outcome indicators and data; and 4. the information needed to assess the risk of bias. The risk of bias in the included studies was assessed using the

```
#1 "Tenofovir"[Mesh]
#2 Tenofovir[Title/Abstract] OR 9-(2-Phosphonylmethoxypropyl)adenine[Title/Abstract] OR 9-PMPA
(tenofovir)[Title/Abstract]      OR      9-(2-Phosphonomethoxypropyl)adenine[Title/Abstract]      OR
9-(2-Phosphonylmethoxypropyl)adenine,            (S)-isomer[Title/Abstract]            OR
9-(2-Phosphonylmethoxypropyl)adenine,    (+-)-isomer[Title/Abstract]   OR   Tenofovir   Disoproxil
Fumarate[Title/Abstract]      OR      Disoproxil      Fumarate,      Tenofovir[Title/Abstract]      OR
Fumarate,   Tenofovir   Disoproxil[Title/Abstract]   OR   Tenofovir   Disoproxil[Title/Abstract])   OR
Disoproxil,   Tenofovir[Title/Abstract]   OR   9-(2-Phosphonylmethoxypropyl)adenine,   (R)-isomer  -
T357098[Title/Abstract]      OR      (R)-9-(2-phosphonylmethoxypropyl)adenine[Title/Abstract]      OR
Viread[Title/Abstract]
#3 #1 OR #2
#4 Entecavir OR Baraclude
#5 "Hepatitis B, Chronic"[Mesh]
#6 Hepatitis B, Chronic[Title/Abstract] OR Chronic Hepatitis B Virus Infection[Title/Abstract] OR
Hepatitis B Virus Infection, Chronic[Title/Abstract] OR Chronic Hepatitis B[Title/Abstract]
#7 #5 OR #6
#8 #3 AND #4 AND #7
```

**Fig 1. PubMed database retrieval strategy.**

RCT bias risk assessment tool recommended in the Cochrane Handbook for Systematic Reviews of Interventions (5.1.0).

## 2.5. Statistical analysis

RevMan 5.3 software was used for the meta-analysis. The dichotomous variables are expressed as the relative risk (RR) as an effect indicator, the continuous variables are expressed as the mean difference (MD) as the effect indicator, and the estimated value and 95% confidence interval (CI) were included as effect analysis statistics. A heterogeneity test was conducted with the results of each study. The fixed effect model was used for the analysis if there was no statistical heterogeneity among the results ($I^2 \leq 50\%$). The sources of heterogeneity needed to be analyzed if there was statistical heterogeneity among the results ($I^2 > 50\%$). After excluding the influence of obvious clinical heterogeneity, the random effects model was used for the analysis. The significance level was set at $\alpha = 0.05$.

## 3. Results

### 3.1. Retrieved results

A total of 3254 studies were initially selected, and 5[10–14] studies were finally included after screening; 4 studies were written in English, and 1 study was written in Chinese. The literature screening process and results are shown in Fig 2.

### 3.2. Basic information of studies

The basic characteristics of the included studies are shown in Table 1, and the bias risk evaluation results are shown in Table 2.

### 3.3. Meta-analysis results

Five studies were included in this study, four in English and one in Chinese. A total of 1,187 individuals were included, including 609 patients who received TDF orally and 578 patients who received ETV orally. This study used medication duration as the basis for subgroups.

**3.3.1. Differences in the probability of normalized ALT indicators.** Four studies reported differences in the probability of normalized ALT indicators between the TDF group and the ETV group. There were 538 patients in the TDF group and 497 patients in the ETV group. A fixed effect model was adopted; at week 24, more patients in the TDF group than in the ETV group had normal ALT levels: $I^2 = 0\%$ [RR = 0.87, 95% CI (0.77, 0.98), $P = 0.02$]; at weeks 96 and 144, there were no statistically significant differences in the probability of normalized ALT indicators between the TDF group and the ETV group: $I^2 = 0\%$ [RR = 0.94, 95% CI (0.88, 1.01), $P = 0.08$], $I^2 = 0\%$ [RR = 0.98, 95% CI (0.92, 1.03), $P = 0.42$] (Fig 3).

**3.3.2. Differences in the probability of negative HBV-DNA results.** Five studies reported differences in the probability of negative HBV-DNA results between the TDF group and the ETV group. There were 609 patients in the TDF group and 578 patients in the ETV group. At week 48, the heterogeneity test result was $I^2 = 87\%$, and the sensitivity analysis suggested that the data from the study by D. Zhang was the main source of heterogeneity. Those data were retained, and the random effects model was adopted, yielding the following results: $I^2 = 87\%$ [RR = 1.08, 95% CI (0.90, 1.30), $P = 0.42$]. Then those data were eliminated, and the fixed effect model was adopted, yielding the following results: $I^2 = 0\%$ [RR = 1.14, 95% CI (1.04, 1.26), $P < 0.01$]. At week 96, more patients in the TDF group seroconverted to become negative for HBV-DNA than in the ETV group ($I^2 = 0\%$) [RR = 1.08, 95% CI (1.03, 1.13), $P < 0.01$]. At week 144, there was no statistically significant difference in the probability of

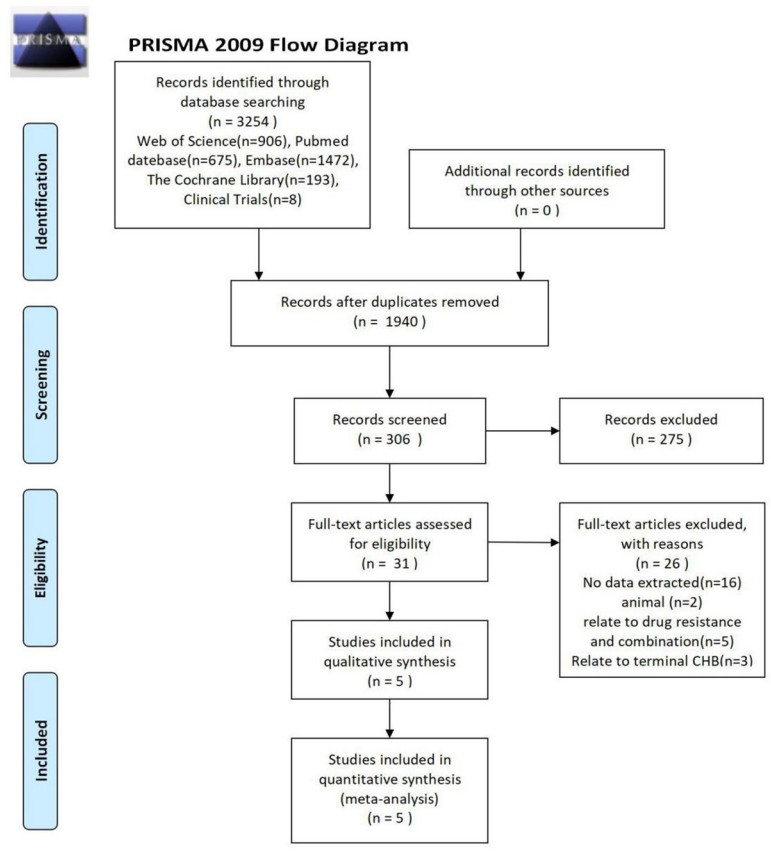

**Fig 2. PRISMA flow diagram of evidence acquisition during the study.**

**Table 1. Basic information of the study.**

| First author | Year | Nation | Type | Drug resistance | Study duration | Sample size TDF | Sample size ETV | Intervention TDF | Intervention ETV | Selection results |
|---|---|---|---|---|---|---|---|---|---|---|
| Dachuan Cai | 2019 | China | RCT | not | 144 weeks | 157 | 158 | Oral administration of TDF 300 mg per day | Oral administration of ETV 0.5 g per day | ①②③④ |
| K. Koike | 2018 | Japan | RCT | not | 24–48 weeks | 109 | 56 | | | ①②④⑤ |
| T. Sriprayoon | 2017 | Thailand | RCT | not | 144 weeks | 200 | 200 | | | ①②③④ |
| D. Zhang | 2017 | China | RCT | not | 48 weeks | 98 | 98 | | | ①②④⑤ |
| Hou-xing Lin | 2016 | China | RCT | not | 96 weeks | 45 | 66 | | | ②③④⑤ |

① Differences in the probability of normalized ALT indicators, ② differences in the probability of HBV-DNA negative results (undetectable), ③ differences in the hepatitis E antigen clearance (HBeAg clearance), ④ differences in the HBeAg seroconversion, and ⑤ differences in the increased probability of creatine kinase (CK).

**Table 2. Bias risk assessments included in the study.**

| Study | | Random sequence generation | Allocation concealment | Blinding method | | Incomplete outcome data | Selective reporting | Other bias |
|---|---|---|---|---|---|---|---|---|
| | | | | Blinding of participants and personnel | Blinding of outcome assessment | | | |
| Dachuan Cai | 2019 | statisticians using the SAS software | Center management | double blind | Blind | drop out or lose 5 patients | unclear | unclear |
| K. Koike | 2018 | unclear | unclear | double blind | unclear | drop out or lose 1 patient | unclear | unclear |
| T. Sriprayoon | 2017 | unclear | unclear | double blind | unclear | drop out or lose 11 patients | high risk | unclear |
| D. Zhang | 2017 | computer generated random sequence | unclear | single blind | unclear | drop out or lose 12 patients | unclear | unclear |
| Hou-xiong Lin | 2016 | unclear | unclear | unclear | unclear | unclear | unclear | unclear |

negative HBV-DNA results between the TDF group and the ETV group ($I^2$ = 0%) [RR = 1.04, 95% CI (1.00, 1.09), $P$ = 0.07] (Fig 4).

**3.3.3. Differences in the probability of HBeAg clearance.** Three studies reported differences in the probability of HBeAg clearance between the TDF group and the ETV group. There were 294 patients in the TDF group and 319 patients in the ETV group. At weeks 48 and 96, there were no statistically significant differences in the probability of HBeAg clearance between the TDF group and the ETV group; the fixed effect model was adopted ($I^2$ = 0%) [RR = 0.97, 95% CI (0.64, 1.47), $P$ = 0.87], ($I^2$ = 0%) [RR = 0.96, 95% CI (0.71, 1.28), $P$ = 0.77]. At week 144, the heterogeneity test yielded an $I^2$ of 87%, and the heterogeneity was likely the result of having too few studies. There was no statistically significant difference in the probability of HBeAg clearance between the TDF group and the ETV group. A random effects model was adopted ($I^2$ = 0%) [RR = 1.05, 95% CI (0.68, 1.62), $P$ = 0.82] (Fig 5).

**3.3.4. Differences in the probability of HBeAg seroconversion.** Five studies reported differences in the probabilities of HBeAg seroconversion between the TDF group and the ETV group. There were 397 patients in the TDF group and 402 patients in the ETV group. At weeks 48 and 96, there were no statistically significant differences in the probability of HBeAg seroconversion between the TDF group and the ETV group; a fixed effect model was adopted

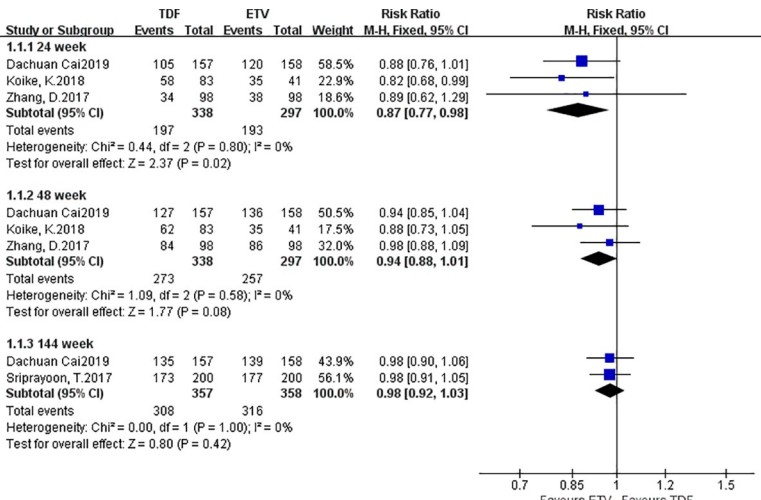

**Fig 3. Forest plot comparing the probability of normalized ALT indicators between TDF and ETV.**

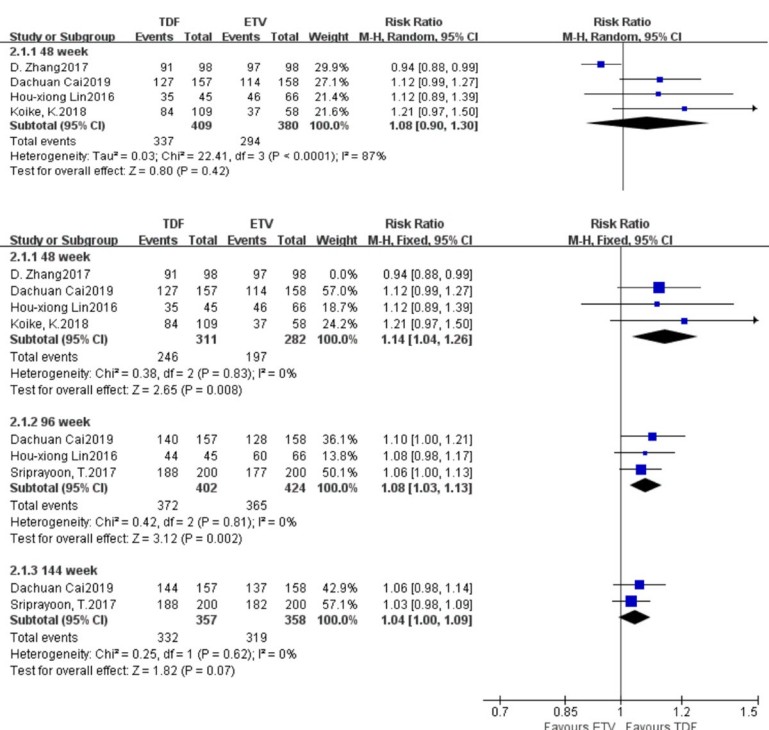

**Fig 4. Forest plot comparing the probability of negative HBV-DNA results between TDF and ETV.**

($I^2$ = 0%) [RR = 0.96, 95% CI (0.63, 1.47), $P$ = 0.85], ($I^2$ = 14%) [RR = 0.79, 95% CI (0.53, 1.16), $P$ = 0.22]. At week 144, the heterogeneity test yielded an $I^2$ of 69%, and the heterogeneity was likely the result of too few studies. There was no statistically significant difference in the probability of HBeAg seroconversion between the TDF group and the ETV group. A random effects model was adopted ($I^2$ = 69%) [RR = 0.93, 95% CI (0.54, 1.61), $P$ = 0.80] (Fig 6).

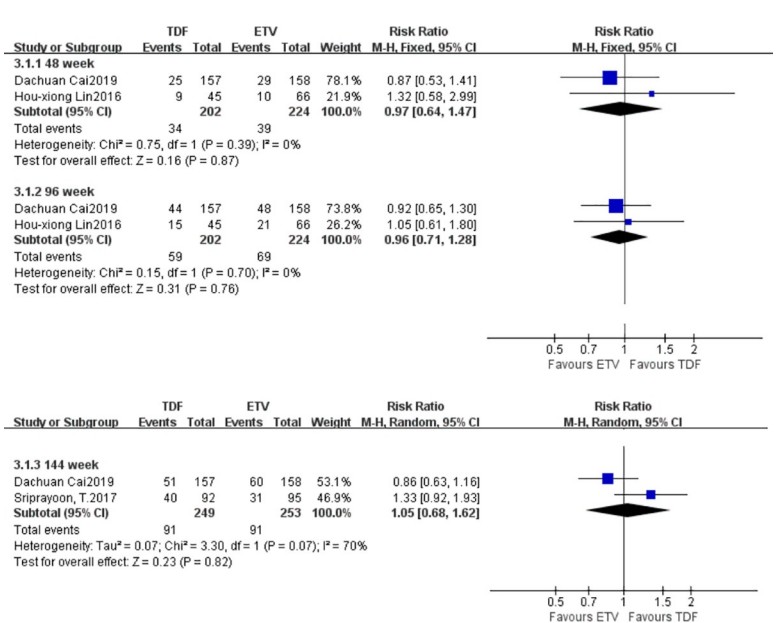

**Fig 5. Forest plot comparing the probability of HBeAg clearance between TDF and ETV.**

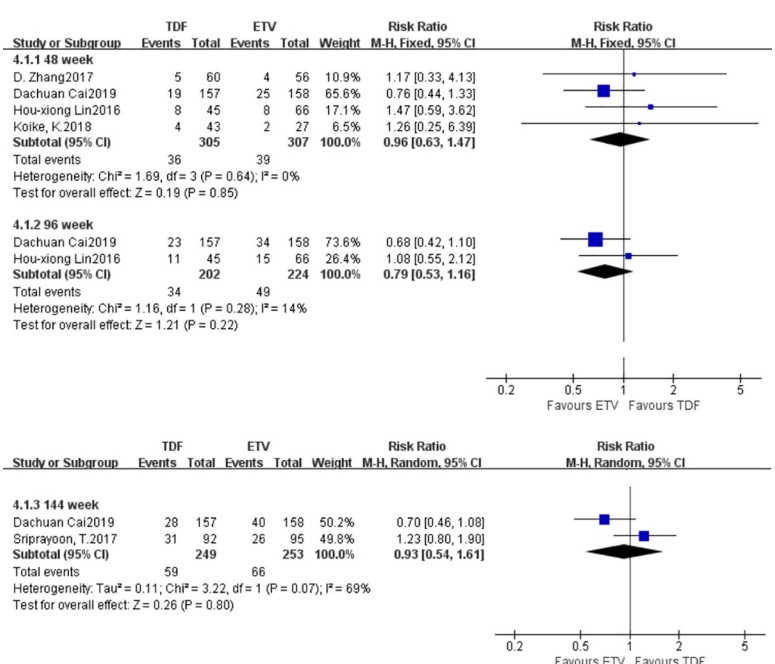

**Fig 6. Forest plot comparing the probability of increased creatine kinase levels between TDF and ETV.**

**3.3.5. Differences in the probability of increased creatine kinase levels.** Three studies reported differences in the probability of increased creatine kinase between the TDF group and the ETV group. There were 214 patients in the TDF group and 178 patients in the ETV group. There was no statistically significant difference in the probability of increased creatine kinase between the TDF group and the ETV group: fixed effect model was adopted, $I^2 = 0\%$ [RR = 0.66, 95% CI (0.27, 1.60), $P = 0.35$] (Fig 7).

## 4. Discussion

TDF and ETV are both first-line treatments for CHB, and their efficacy and safety are widely recognized[15, 16]. However, it is difficult to choose between TDF and ETV for patients who are initially diagnosed with CHB.

The treatment of CHB is usually considered a clinical cure, which refers to a sustained virological response with negative HBsAg or positive HBsAb transformation, normal ALT levels, and mild or no lesions in the liver tissue[17]. Therefore, HBV-DNA conversion and ALT normalization were selected as indicators. The normal value of ALT varies from person to person, region to region, or device to device. In the analysis of the data included in this study, no special requirements were imposed for the normalcy of ALT. Similarly, different studies have different methods for measuring HBV-DNA and different units for the measurement results. The

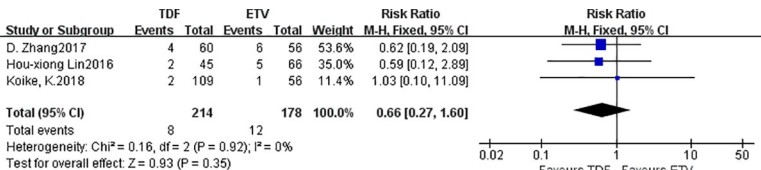

**Fig 7. Forest plot comparing the probability of HBeAg seroconversion between TDF and ETV.**

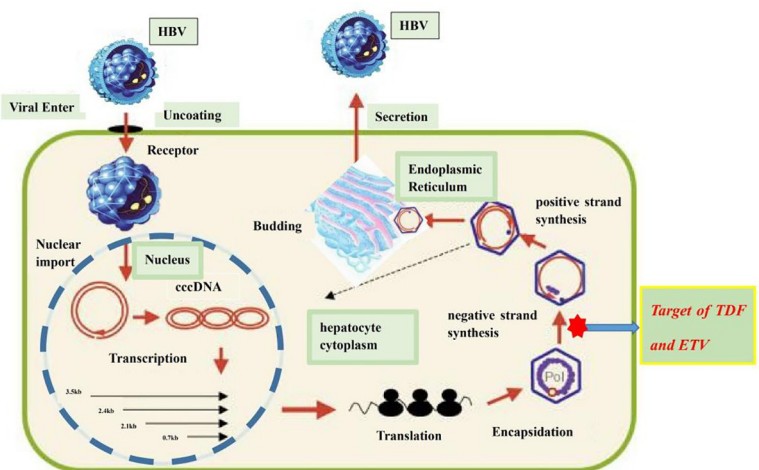

**Fig 8. The mechanism of TDF and ETV anti-HBV.**

units of HBV-DNA measurement are converted as follows: 1 IU/mL is approximately equal to 5–6 copies/mL. Therefore, the lower limit of quantitation (LLQ) of each experiment is similar. However, when different approaches are used to determine the HBV-DNA level, then extremely low HBV levels in the blood cannot be measured. This study will not impose special requirements for measuring HBV-DNA.

Tenofovir is a nucleotide reverse transcriptase inhibitor that inhibits reverse transcriptase in a similar way to nucleoside reverse transcriptase inhibitors and thus has potential anti-HBV activity[18]. Tenofovir bisphosphonates, the active component of tenofovir, inhibit the viral polymerase by directly competing with the natural deoxyribose substrates and terminating DNA strands by inserting DNA. Entecavir is a guanine nucleoside analogue, and its antiviral pharmacological action is similar to that of tenofovir[19]. (Fig 8)

By comparison, tenofovir was found to have an advantage in inhibiting the virus in the early stage (week 96), and entecavir was superior in protecting liver function in the early stage (week 24). However, the difference between tenofovir and entecavir in inhibiting the virus and protecting liver function gradually decreased with the increased duration of treatment, and the difference was not statistically significant by week 144. Koike[11] also suggested that a correlation between ALT and HBsAg may exist and is important in the HBsAg reduction process. Higher ALT in the early stage seems to indicate lower HBV in the later stage.

The reason why TDF is better at suppressing the virus in the early stages of CHB but is less protective of liver function than ETV remains unclear. First, all included studies suggested that both ETV and TDF have strong antiviral and protective liver function effects. These differences are only the differences between TDF and ETV. The reasons for such discrepancies might include the following: (1) a high immune response could suppress the hepatitis B virus but may also damage liver cells; (2) drug metabolites influence liver cells, and ETV and TDF are antiviral drugs but not liver-protecting drugs; thus, although most of their metabolites are excreted through the kidneys, they all target the liver cells and may have different effects on the liver cells while clearing the virus; and (3) bias or other reasons could cause deviations; thus, because meta-analysis collates scattered data for analysis, small effects could be magnified to produce meaningful results. Fewer trials were included in this study, and the possibility of bias was relatively large. It is hoped that by studying the differences between TDF and ETV, more effective and safe antiviral drugs can be developed.

HBeAg clearance or HBeAg seroconversion can be considered indications that CHB is under control. In China, that is referred to as the "big three positives (Hepatitis B HBeAg)" turning to negative or to "small three positive"[20]. Elevated HBeAg levels indicate that the patient is highly infectious and usually appears in the early or active stages of HBV infection [21]. Therefore, the clearance probability and seroconversion probability of HBeAg were selected as indicators in this study. There were no statistically significant differences in the clearance probability and seroconversion probability of tenofovir and entecavir for HBeAg.

As for the adverse reactions to oral tenofovir and entecavir, the data are scattered and lack systematic elaboration and analysis; only the difference in the probability of CK level increase was analyzed. The results suggested that there was no statistically significant difference in the probability of CK level increase between TDF and ETV. In the literature, the discussed adverse reactions were mostly well tolerated. In this study, the adverse reactions in the TDF and ETV groups were considered mild and suitable for the long-term oral treatment of CHB[22, 23].

Because this study included RCTs only, some important outcome measures were not included, for example, hepatocellular carcinoma (HCC), mortality, transplantation, etc. The results of these studies require long-term observational studies, so cohort studies or case control studies are usually used. Zhang[14] found that compared with ETV, oral TDF carried a lower risk of HCC, but there were no statistically significant differences in mortality or transplantation. A meta-analysis of cohort studies also suggested that there was a better effect of TDF in reducing HCC incidence than ETV, while there was no significant difference in incidence of death or transplantation, encephalopathy or variceal bleeding between the two groups[24].

In this study, we searched for studies in Chinese and English. Our team can only access articles in these two languages, which represents a limitation of our team. In China, there are more than 100 million hepatitis b carriers and tens of millions of CHB patients. Chinese researchers have attached great importance to chronic hepatitis b, and the level of research on CHB in China is recognized worldwide[25].

The limitations of this meta-analysis are as follow:

1. The number of retrieved RCT studies was too small, and too much weight was accounted for by the analysis of Dachuan Cai et al.; moreover, the outcome indicators adopted in this study are not sufficiently novel, although new information will be provided in future updates.

2. The regions included in the study were all in Asia, and whether the results are universal needs to be further demonstrated.

3. The data regarding adverse events are relatively scattered, and it was only possible to analyze the probability of CK level increase.

Among the studies included in the study, the study by Hou-xiong Lin concluded that TDF was more efficacious than ETV, while the other 4 studies all concluded that TDF and ETV had similar therapeutic effects and adverse reactions. This study suggests that TDF has a greater ability to inhibit the virus in the early stage; ETV has a small effect on the liver in the early stage, but its therapeutic effect is similar in the long term (144 weeks), with a small incidence of adverse reactions, which were mostly tolerable and did not affect the long-term oral treatment of CHB. While patients choose drugs after receiving an initial diagnosis, they can compare characteristics other than the efficacy of the two drugs, such as the generation of long-term oral drug resistance, the price, the convenience of regular review, the habit of taking drugs, the risk of hepatocellular carcinoma (HCC), and mortality. Furthermore, TDF has an

advantage in the incidence of HCC. TDF and ETV need to be studied in more RCTs and a large prospective cohort study.

## Supporting information

**S1 Checklist. PRISMA checklist.**
(DOCX)

## Acknowledgments

At the point of finishing this paper, I would like to express my sincere gratitude to all those who have lent me assistance in the course of writing this paper. I would like to express my gratitude to my workmates who offered me references and information in a timely manner. Then, I would like to thank the leaders, teachers and staff, especially at my alma mater, Nanjing Medical University. Without their help, it would have been much harder for me to finish my course of study and this paper.

## Author Contributions

**Conceptualization:** Mao-bing Chen, Hua Wang, Qi-han Zheng, Jin-nuo Fan, Yun-long Ding, Jia-li Niu.

**Data curation:** Mao-bing Chen, Xu-wen Zheng, Yun-long Ding, Jia-li Niu.

**Formal analysis:** Mao-bing Chen.

**Investigation:** Hua Wang, Xu-wen Zheng, Jin-nuo Fan, Yun-long Ding, Jia-li Niu.

**Methodology:** Mao-bing Chen, Xu-wen Zheng.

**Resources:** Mao-bing Chen, Jin-nuo Fan.

**Software:** Mao-bing Chen.

**Supervision:** Hua Wang, Qi-han Zheng.

**Writing – original draft:** Mao-bing Chen, Hua Wang, Xu-wen Zheng, Jin-nuo Fan, Yun-long Ding, Jia-li Niu.

**Writing – review & editing:** Mao-bing Chen.

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
