## [Decision Letter · Decision Letter 0]

3 Sep 2019

PONE-D-19-21194

Effectiveness of tenofovir and entecavir in nucleos(t)ide analogue-naive chronic hepatitis B

PLOS ONE

Dear Mr Chen,

Thank you for submitting your manuscript to PLOS ONE. After careful consideration, we feel that it has merit but does not fully meet PLOS ONE’s publication criteria as it currently stands. Therefore, we invite you to submit a revised version of the manuscript that addresses the points raised during the review process.

We would appreciate receiving your revised manuscript by Oct 18 2019 11:59PM. To enhance the reproducibility of your results, we recommend that if applicable you deposit your laboratory protocols in protocols.io, where a protocol can be assigned its own identifier (DOI) such that it can be cited independently in the future. For instructions see: http://journals.plos.org/plosone/s/submission-guidelines#loc-laboratory-protocols

We look forward to receiving your revised manuscript.

Kind regards,

Seung Up Kim

Academic Editor

PLOS ONE

Journal Requirements:

2. Please include your tables as part of your main manuscript and remove the individual files. Please note that supplementary tables (should remain/ be uploaded) as separate "supporting information" files

'This work was funded by the Guidance plan for social development of

Changzhou Municipal Science and Technology (CE20175008), Changzhou City, Jiangsu Province,P.R.

China. The funder had no role in the study design, data collection, data analysis, data interpretation,

writing of the report, decision to publish, or preparation of the manuscript.'

'X'

'X'

5. Please amend either the title on the online submission form (via Edit Submission) or the title in the manuscript so that they are identical.

Additional Editor Comments (if provided):

Reviewers' comments:

Reviewer's Responses to Questions

**Comments to the Author**

1. Is the manuscript technically sound, and do the data support the conclusions?

Reviewer #1: Yes

Reviewer #2: No

Reviewer #3: Yes

2. Has the statistical analysis been performed appropriately and rigorously? 

Reviewer #1: Yes

Reviewer #2: I Don't Know

Reviewer #3: No

3. Have the authors made all data underlying the findings in their manuscript fully available?

Reviewer #1: Yes

Reviewer #2: No

Reviewer #3: Yes

4. Is the manuscript presented in an intelligible fashion and written in standard English?

Reviewer #1: Yes

Reviewer #2: No

Reviewer #3: Yes

5. Review Comments to the Author

Reviewer #1: There is only minor comment. Since entecavir was prescribed earlier compared to tenofovir, authors should perform a subgroup analysis focusing studies with similar follow-up treatment duration.

Reviewer #2: 1. There are few new information in the present manuscript. The main goals of antiviral therapy in chronic hepatitis B patients are decreasing liver related mortality and morbidity by suppressing viral replication. Therefore, the primary goal of the manuscript should be focused in achieving these goals, not the percentage of ALT normalization nor HBV reduction, which are already well-known. Comparing the antiviral effect between entecavir and tenofovir cannot make any significant new finding in the clinical aspect.

Reviewer #3: This study (PONE-D-19-21194) was a systemic review and meta-analysis to compare effectiveness of tenofovir and entecavir.

1. The authors suggested that tenofovir was more effective than entecavir in inhibiting the virus in the early stage, while entecavir was more effective than tenofovir in protecting liver function in the early stage. However, we all know that improvement of liver function follows suppression of HBV replication. Therefore, please discuss why early HBV suppression by tenofovir could not improve the liver function in the early stage.

2. This study compared the HBV DNA suppression and ALT normalization between entecavir and tenofovir. However, there are discrepancies in the definitions of HBV DNA suppression and ALT normalization among studies as follows. How the authors treat these discrepancies in their analysis?

Ref No ULN of ALT Lower limit of HBV DNA detection

8 40 IU/L 20 IU/mL

9 31 IU/L 2.1 log10 copies/mL

10 Not mentioned 20 IU/mL

11 Journal search failed Journal search failed

12 Not mentioned Not mentioned

3. One study (Ref No 11) could be searched only by “China Knowledge Resource Integrated Database”. Therefore, this study could not be reviewed by all peoples except Chinese. I think this study should be excluded in the meta-analysis to improve the clearness of the study.

6. PLOS authors have the option to publish the peer review history of their article (what does this mean?). If published, this will include your full peer review and any attached files.

Reviewer #1: No

Reviewer #2: No

Reviewer #3: No

---

## [Author Response · Author response to Decision Letter 0]

17 Sep 2019

A point-by-point response 

Reviewer #1: There is only minor comment. Since entecavir was prescribed earlier compared to tenofovir, authors should perform a subgroup analysis focusing studies with similar follow-up treatment duration.

Response: We highly appreciate your valuable comments. We performed subgroup analyses for different follow-up times. Accordingly, the abstract, results, and discussion have been revised. These changes did not influence the main content of the paper.

Reviewer #2: 1. There are few new information in the present manuscript. The main goals of antiviral therapy in chronic hepatitis B patients are decreasing liver related mortality and morbidity by suppressing viral replication. Therefore, the primary goal of the manuscript should be focused in achieving these goals, not the percentage of ALT normalization nor HBV reduction, which are already well-known. Comparing the antiviral effect between entecavir and tenofovir cannot make any significant new finding in the clinical aspect.

Response: Thank you for this suggestion. The article is a meta-analysis, which is a quadratic analysis of the literature. Although we would have liked to introduce novel techniques and methods, adopting such novel techniques and methods is difficult if corresponding studies have not been performed or only a single study is available in the literature. Generally, more data can be obtained by adopting more widely used techniques or methods, and more data can make the results of the meta-analysis more reliable. Very novel techniques and methods that have only been reported in one study would not be of significance for a meta-analysis. We will monitor such research and update the meta-analysis if more novel articles are published. These details will be generalized in the limitations section of this study.

Reviewer #3: This study (PONE-D-19-21194) was a systemic review and meta-analysis to compare effectiveness of tenofovir and entecavir.

1. The authors suggested that tenofovir was more effective than entecavir in inhibiting the virus in the early stage, while entecavir was more effective than tenofovir in protecting liver function in the early stage. However, we all know that improvement of liver function follows suppression of HBV replication. Therefore, please discuss why early HBV suppression by tenofovir could not improve the liver function in the early stage.

2. This study compared the HBV DNA suppression and ALT normalization between entecavir and tenofovir. However, there are discrepancies in the definitions of HBV DNA suppression and ALT normalization among studies as follows. How the authors treat these discrepancies in their analysis?

Ref No ULN of ALT Lower limit of HBV DNA detection

8 40 IU/L 20 IU/mL

9 31 IU/L 2.1 log10 copies/mL

10 Not mentioned 20 IU/mL

11 Journal search failed Journal search failed

12 Not mentioned Not mentioned

3.One study (Ref No 11) could be searched only by “China Knowledge Resource Integrated Database”. Therefore, this study could not be reviewed by all peoples except Chinese. I think this study should be excluded in the meta-analysis to improve the clearness of the study.

Response: Thank you for these suggestions. 

Why tenofovir is better at suppressing the virus in the early stages of CHB but less protective of liver function than entecavir will be mentioned in the discussion. Few specific basic studies have focused on the differences between oral TDF and ETV, and the answer can only be found through clinical practice and a survey of the literature. First, both ETV and TDF have strong antiviral and liver function improvement effects, and the possible reasons for these effects include the following: (1) a high immune response can damage liver cells but also suppresses the hepatitis B virus; (2) drug metabolites damage liver cells, and ETV and TDF are antiviral drugs and not liver-protecting drugs; thus, although most of their metabolites are excreted through the kidneys, they all target the liver and may have different effects on the liver while clearing the virus; (3) Bias or other reasons may cause deviations, and because meta-analyses collate scattered data for analysis, small effects may be magnified to produce meaningful results. These possibilities are hypothetical; however, if the reasons for these effects can be worked out, antiviral drugs can be effective in protecting liver function while providing high antiviral activity.

Reviewer #3 provided a careful review and identified discrepancies in the definitions. The meta-analysis required that the included studies be similar but also present differences that could not be absolutely unified. For example, different analyzers define the normal value of ALT differently and different analytical methods measure different HBV thresholds. Therefore, the normal amount of ALT was not limited. We believe that the use of different ALT values in the literature used in the meta-analysis will not affect the reliability of this study. The conversion between units of HBV-DNA measurement was as follows: 1 IU/mL is approximately equal to 5-6 copies/mL. Therefore, the lower limit of quantitation (LLQ) of each experiment is similar. These heterogeneities between studies can be discussed in this meta-analysis.

Hou-xiong Lin’s study is in Chinese. In the Methods section of our study, we stated that Chinese literature and English literature will be included. We agree that there are many low-quality papers in the Chinese literature. When performing research, we usually do not search the CNKI database because the quality of some research is unacceptable. However, China has more than one hundred million chronic hepatitis b carriers. Chinese researchers attach great importance to the study of CHB and the level of research on CHB in China is acceptable. Therefore, we do not believe that this research should be left out. The basic requirement of the meta-analysis is to be comprehensive, so this information will be added to the discussion. The reliability of the research will be further analyzed; moreover, we will remove this study and determine whether the results are affected.

---

## [Decision Letter · Decision Letter 1]

7 Oct 2019

PONE-D-19-21194R1

Comparative efficacy of tenofovir and entecavir in nucleos(t)ide analogue-naive chronic hepatitis B: A systematic review and meta-analysis

PLOS ONE

Dear Mr Chen,

Thank you for submitting your manuscript to PLOS ONE. After careful consideration, we feel that it has merit but does not fully meet PLOS ONE’s publication criteria as it currently stands. Therefore, we invite you to submit a revised version of the manuscript that addresses the points raised during the review process.

We would appreciate receiving your revised manuscript by Nov 21 2019 11:59PM. To enhance the reproducibility of your results, we recommend that if applicable you deposit your laboratory protocols in protocols.io, where a protocol can be assigned its own identifier (DOI) such that it can be cited independently in the future. For instructions see: http://journals.plos.org/plosone/s/submission-guidelines#loc-laboratory-protocols

We look forward to receiving your revised manuscript.

Kind regards,

Seung Up Kim

Academic Editor

PLOS ONE

Additional Editor Comments (if provided):

Congratulations on your nice work.

As a handling editor, I'd like to give some more minor comments.

1. As the reviewer indicated, several statistical description seems missed.

2. I also believe that there might be no difference between ETV vs. TDF in long-term prognosis. However, as you know, some recent studies have shown the superiority of TDF over ETV, probably due to many biased selection of these two drugs. Please discuss and focus more on the potential bias that have made the different efficacy of these two drugs in several recent studies, not to provide potentially wrong signals to physicians. You may refer the related editorials to upgrade Discussion section.

Reviewers' comments:

Reviewer's Responses to Questions

**Comments to the Author**

1. If the authors have adequately addressed your comments raised in a previous round of review and you feel that this manuscript is now acceptable for publication, you may indicate that here to bypass the “Comments to the Author” section, enter your conflict of interest statement in the “Confidential to Editor” section, and submit your "Accept" recommendation.

Reviewer #1: All comments have been addressed

Reviewer #2: (No Response)

Reviewer #3: All comments have been addressed

2. Is the manuscript technically sound, and do the data support the conclusions?

Reviewer #1: Yes

Reviewer #2: No

Reviewer #3: Yes

3. Has the statistical analysis been performed appropriately and rigorously? 

Reviewer #1: Yes

Reviewer #2: I Don't Know

Reviewer #3: Yes

4. Have the authors made all data underlying the findings in their manuscript fully available?

Reviewer #1: Yes

Reviewer #2: (No Response)

Reviewer #3: Yes

5. Is the manuscript presented in an intelligible fashion and written in standard English?

Reviewer #1: Yes

Reviewer #2: (No Response)

Reviewer #3: Yes

6. Review Comments to the Author

Reviewer #1: (No Response)

Reviewer #2: It is well known by the key trials that tenofovir is more potent in inhibiting viral replication than entecavir while ALT normalization is less prominent. The manuscript does not show more information by meta-analysis. Moreover, the present study abstracted data performed in small number of patients in Asia. I recommend the authors to collect data from large key studies of each drug and make analysis how tenoforvir and entecavir affected the treatment outcomes. Please, refer to biostatistics specialist for the analysis. Last, please provide how you manage the biases of each study (e,g QUIPS).

Reviewer #3: I think that the authors made appropriate corrections and good responses to my comments. I have no more comments.

7. PLOS authors have the option to publish the peer review history of their article (what does this mean?). If published, this will include your full peer review and any attached files.

Reviewer #1: No

Reviewer #2: No

Reviewer #3: No

---

## [Author Response · Author response to Decision Letter 1]

18 Oct 2019

A point-by-point response 

Reviewer #2: It is well known by the key trials that tenofovir is more potent in inhibiting viral replication than entecavir while ALT normalization is less prominent. The manuscript does not show more information by meta-analysis. Moreover, the present study abstracted data performed in small number of patients in Asia. I recommend the authors to collect data from large key studies of each drug and make analysis how tenoforvir and entecavir affected the treatment outcomes. Please, refer to biostatistics specialist for the analysis. Last, please provide how you manage the biases of each study (e,g QUIPS).

Response: We highly appreciate your valuable comments. I have studied the relevant issues carefully and revised my paper.

1.In the discussion section, I analyzed the differences between ALT and HBV-DNA and tried to give possible reasons. These reasons are only speculation. TDF and ETV cause differences between ALT and HBV-DNA, and a mechanism has not been described in the relevant literature. I think there is a potential relationship between ALT and HBV-DNA. As mentioned in Koike’s study, elevated ALT in the early stage might indicate decreased HBV DNA in the later stage, and the prognosis may be better. This was mentioned as a phenomenon in the study. Thank you for your reminder; we further elaborated on their relationship.

2.I want to explain for myself. When I searched the literature, I did not put any restrictions, such as area, race, etc., on the literature. Instead, I wanted to search the literature from different regions. Perhaps more can be drawn from the different regions and races. However, all of the studies were conducted in Asia. It is possible that the incidence of HBV is high in Asia, and relevant studies are abundant. I also hope that there will be more studies in the future, and I will update my meta-analysis at that time. With regard to the inclusion of other outcome indicators, we have increased our discussion in this regard, and we will illustrate the incidence of HCC, death and transplantation in the systematic review.

3.In meta-analysis, we usually use the methods recommended in the Cochrane Handbook to evaluate risk bias. This method is specific to RCTs. I have carried out Bias Risk Assessments. In addition, I tried to perform QUIPS as well (Table 1). I will upload this form as an attachment.

Table 1. Risk of bias/quality scores.

Study Bias field

 Study Participation Study Attrition Risk Factor Measurement Outcome Measurement Study Confounding Statistical Analysis and Reporting

Dachuan Cai 2019 low low Moderate low moderate moderate

Koike, K. 2018 low low low low moderate low

Sriprayoon, T. 2017 low low low low low low

D.Zhang 2017 low low low low moderate moderate

Hou-xing Lin 2016 low moderate moderate low moderate low

---

## [Editor Report · Decision Letter 2]

22 Oct 2019

Comparative efficacy of tenofovir and entecavir in nucleos(t)ide analogue-naive chronic hepatitis B: A systematic review and meta-analysis

PONE-D-19-21194R2

Dear Dr. Chen,

We are pleased to inform you that your manuscript has been judged scientifically suitable for publication and will be formally accepted for publication once it complies with all outstanding technical requirements.

With kind regards,

Seung Up Kim

Academic Editor

PLOS ONE

Additional Editor Comments (optional):

Congratulation on your nice work.
---

## [Editor Report · Acceptance letter]

12 Nov 2019

PONE-D-19-21194R2 

Comparative efficacy of tenofovir and entecavir in nucleos(t)ide analogue-naive chronic hepatitis B: A systematic review and meta-analysis 

Dear Dr. Chen:

I am pleased to inform you that your manuscript has been deemed suitable for publication in PLOS ONE. Congratulations! Your manuscript is now with our production department. 

With kind regards,

on behalf of

Dr. Seung Up Kim 

Academic Editor

PLOS ONE